# GLP-1 RA Prescribing Errors in a Multidisciplinary Digital Weight-Loss Service: A Retrospective Quantitative Analysis

**DOI:** 10.3390/healthcare12202093

**Published:** 2024-10-21

**Authors:** Louis Talay, Matt Vickers, Sarah Fuller

**Affiliations:** 1Faculty of Arts and Social Sciences, University of Sydney, Sydney, NSW 2050, Australia; 2Eucalyptus, Sydney, NSW 2000, Australia; matt@eucalyptus.vc (M.V.); sarah.fuller@eucalyptus.vc (S.F.)

**Keywords:** prescribing errors, digital health, chronic disease, electronic prescribing, safety, digital obesity program, GLP-1 RA

## Abstract

Background: Digital weight loss services (DWLSs) that use Glucagon-like peptide-1 receptor agonists (GLP-1 RAs) have demonstrated potential in contributing to a shift in global obesity rates. However, reasonable concerns have been raised about the prescribing safety of these services. Prior to this study, electronic prescribing safety had only been investigated in hospital settings and community clinics. Methods: This study retrospectively analyzed prescribing errors committed over a 6-month period in 2023 at Australia’s largest GLP-1 RA-supported DWLS. Results: The analysis found that 1654 (4.4%) of the 37323 audited GLP-1 RA prescriptions contained an error. Most errors pertained to insufficient safety counselling (49.15%) and inadequate investigations of potential contraindications (30.29%). Although a large portion of prescribing errors were detected via the automated query method (64.9%), the other three auditing methods all detected a significant number of true errors (>100). Patients from the highest body mass index category (40+ kg/m^2^) were overrepresented in the service’s prescribing error data. Conclusions: These findings lay a vital foundation in the emerging literature on GLP-1 RA-supported DWLSs.

## 1. Introduction

Obesity is arguably the most concerning public health issue in the modern world. The most recent global analysis revealed that two and a half billion adults (43% of the global population) were overweight, 890 million (16%) of whom live with obesity [1]. In Australia, these figures are even more alarming, with two-thirds (66%) of people over 18 considered overweight, of whom 32% live with obesity [2]. Both international and Australian obesity rates have risen steadily since the 1990s [2,3].

To combat these trends, major health institutions emphasize the importance of continuous multidisciplinary care [4,5]. However, accessing this level of care in face-to-face settings on a mid- to long-term basis has historically proven difficult for patients with significant family or work commitments [6,7]. In countries with substantial regional populations, these challenges are often compounded by a geographical barrier to obesity care [8].

Digital weight-loss services (DWLSs) have emerged as a potential solution to this access problem, allowing patients to attend consultations with multidisciplinary teams (MDTs) at a time and location of their convenience. Many DWLSs utilize Glucagon-like peptide-1 receptor agonists (GLP-1 RAs), either as standalone treatment, or as a supplement to lifestyle coaching [9]. Although randomized controlled trials (RCTs) have consistently evidenced the safety and efficacy of GLP-1 RAs in clinical settings [10,11,12], stakeholders remain concerned about their use in real-world DWLSs [9,13]. Key concerns are that many DWLSs do not follow international guidelines in using GLP-1 RA treatment as a supplement to continuous lifestyle therapy and that they allow unsuitable patients to obtain such medications [13,14]. At present, both concerns are very reasonable. While the DWLS spectrum is broad and contains several providers who offer GLP-1 RAs strictly as a supplement to lifestyle therapy [7,15], many services still appear to deliver GLP-1 RA prescriptions with little to no follow-up care [9,14]. In regard to allowing unsuitable patients to obtain the medications, problems of this nature ultimately stem from one of two error types in DWLS care models: pharmacy dispensing errors or prescribing errors [16].

A recent study found a dispensing error rate of 0.35% in Australia’s largest GLP-1 RA-supported DWLS [16]. However, this outcome was difficult to evaluate given the absence of comparable data. Research relevant to DWLS prescribing errors is comparably scarce. While there is a good amount of evidence to suggest that digital prescribing models deliver better safety outcomes than hand-written alternatives due to their facilitated incorporation of data analytics [17,18,19], little is known about GLP-1 RA prescribing errors, or errors specific to DWLSs. To the knowledge of the investigators, the only study relevant to either area was a 2024 short communication on three cases of Semaglutide administration errors among Type 2 diabetes patients [20]. In terms of Australian digital prescribing benchmarks, the most recent evidence appears to come from a 2022 time-series study at a secondary hospital setting, which reported a prescribing error rate of 27% among the 13,841 e-prescriptions that were reviewed [18]. A 2012 before and after study at two Australian teaching hospitals reported prescribing error rates between 10.2 and 17.3 percent in the e-prescribing cohorts (versus 39.4 to 51.6 in the paper-based prescribing cohorts) [21]. Earlier studies in Australian hospital settings had observed e-prescribing error rates ranging from 9 to 20 percent [22]. A 2013 study commissioned by the Australian government estimated that the annual cost of medication-related hospital admissions was 1.2 billion Australian dollars and that most errors that led to these admissions were preventable [13].

Regarding community settings, data on digital prescribing error rates appear to be limited to the United States and Europe, with figures ranging from 6.6 to 51.4 percent [23,24,25]. This error rate variability largely stems from the error detection method. Most studies to date have relied on retrospective reviews of prescriptions by clinicians [23], yet some opt for computerized detection [24], or patient-reported adverse events [20]. Given that certain prescribing errors do not incentivize patient error reporting (e.g., higher doses of a drug that the perceive to deliver favorable outcomes), the first two methods tend to result in significantly higher error rates. The most common prescribing errors in digital community settings are typically those associated with incorrect dose or frequency, incorrect drug concentration or strength, and incorrect medication [26]. Findings from a 2019 systematic review suggest that ongoing advances in electronic prescribing technology have increased the magnitude of dosing errors and adverse event reductions [25].

The significance of prescribing safety in a global public health context is arguably best evidenced by the World Health Organization’s 2017 ‘Medication Without Harm’ patient safety challenge—only the third challenge of the organization’s history [27]. The challenge’s goal of “reducing patient harm generated by unsafe medication practices and medication errors” has not been adequately addressed in Australia. A 2024 government report highlighted the failure of the current Australian health system to collect sufficient medication safety data, introduce industry-wide standards, and implement appropriate controls for direct-to-consumer communications, among other things [28]. These findings are a major concern for the increasingly large number of Australians who are using GLP-1 RA-supported DWLSs [14].

This study aims to analyze the prescribing error rate in Australia’s largest GLP-1 RA-supported DWLS, Eucalyptus. In doing so, it seeks to gain an understanding of the type and severity of errors committed through the service. It is believed that these findings will generate vital foundational knowledge for the emerging fields of digital prescribing safety and digital obesity care.

## 2. Materials and Methods

### 2.1. Study Design

This study retrospectively analyzed a dataset of GLP-1 RA orders received by patients of the Eucalyptus DWLS (Juniper for women; Pilot for men) between 1 April and 1 October 2023. This design was adopted in accordance with the NHS Health Research Authority’s ‘Defining Research’ matrix [29], aligning with the following criteria: “designed and conducted solely to define or judge current care or service”; “measures current service without reference to a standard”; “involves analysis of existing data”; and “patients have chosen intervention independently of the service evaluation”. The Bellberry Human Ethics Committee approved the study on 22 November 2023 (No. 2023-05-563-A-1). All patients consented to the publication of their de-identified data in this research. Key ethical implications of the study were the potential for patient harm if prescribing errors were not addressed by Eucalyptus in a timely manner or if patient data were not adequately deidentified and secured. An assessment of clinical responses to prescribing errors was considered to be beyond the scope of this study. Eucalyptus accepts full responsibility for any GLP-1 RA prescribing errors from its DWLS that lead to patient harm.

### 2.2. Program Overview

The Eucalyptus DWLS is accredited through the Australian Council on Healthcare Standards [30]. The service has only ever provided GLP-1 RA-supported therapy, i.e., it has never offered standalone lifestyle or GLP-1 RA treatment. All patients are allocated an MDT consisting of a physician, a university-qualified health coach, a pharmacist, and a medical support officer to guide them through personalized lifestyle coaching and GLP-1 RA therapy. All patient-MDT communication is conducted via the Juniper and Pilot online platforms and is automatically uploaded to the Eucalyptus central data repository on Metabase—an open-source business intelligence tool. Lifestyle coaching includes access to multimodal educational materials, progress trackers, and meal and exercise plans. Patients can request changes to their personalized lifestyle plans at any stage of their care journey.

To access the service, prospective patients complete an online pre-consultation questionnaire, which contains up to 100 questions. A doctor or nurse practitioner reviews these responses and often solicits additional information such as blood test results, medical imaging, and reports from previous clinicians. Once they have obtained sufficient information, they determine a patient’s eligibility for the Eucalyptus DWLS. Eligibility decisions are largely based on GLP-1 RA product information documents that detail body-mass index (BMI) ranges, contraindications and drug interactions [31,32]. BMI cutoffs for the GLP-1 RAs that were used during the study period were 27 kg/m^2^ for patients with at least one weight-related comorbidity (e.g., symptomatic cardiovascular disease, sleep apnea) or patients of non-Caucasian ethnicity, and 30 kg/m^2^ for everyone else. Contraindications include multiple endocrine neoplasia syndrome type 2; a family history of medullary thyroid carcinoma; diabetic retinopathy complications; pancreatitis; hypoglycemia and concomitant insulin use; a previous acute kidney injury; or a known hypersensitivity to GLP-1 RA product components.

Upon payment of their first monthly subscription fee, patients receive a prescription for a 4-month supply of GLP-1 RA medications. Patients are required to attend a follow-up consultation with their prescribing doctor or nurse practitioner during month 4 to determine their suitability for ongoing treatment (and thus another prescription for a 4-month supply of GLP-1 RAs). Thereafter, follow-up consultations continue to be held at 4-month intervals. Ad hoc consultations can be requested by patients or their MDT at any stage of a patient’s care journey. As is the case for MDT–patient communication, all prescribing decisions are automatically uploaded to the Eucalyptus central data repository.

The Eucalyptus clinical auditing team—consisting of registered doctors and pharmacists—implements data analytics in the service’s central data repository to monitor prescribing decisions. The team uses a series of Metabase queries to detect any prescriber decision that resembles a high-risk error, such as the decision to not request further information about a patient’s cancer diagnosis, or the premature escalation of a patient’s GLP-1 RA dose. These queries pull such data onto thematic dashboards that the auditing team manually audits every 24–72 h (the highest risk categories are reviewed every 24 h). A high percentage of errors detected through these queries end up being flagged as non-errors by Eucalyptus auditors. The reason for this is that the queries do not capture follow-up question input. For example, a patient might select “insulin” or “gallstones” (both potential high-risk contraindications) in one or more relevant questions but confirm in personalized follow-up questions that they are no longer on insulin and/or already had their gallbladder removed. In such cases, a Eucalyptus auditor will remove the prescription’s error tag. In cases of confirmed errors, the responsible auditor will immediately inform the patient’s MDT to determine the appropriate course of intervention. An example of an automated query for a high-risk contraindication dashboard (symptomatic cardiovascular disease) is provided in Figure 1.

In addition to this automated query protocol, the company conducts 3 other types of manual audits. These include ad hoc audits, conducted in response to internal or external insights of misprescription; random audits, which run at a frequency consistent with a 95 percent confidence interval of the entire consultation sample; and new prescriber audits, completed for the first 100 consultations of any new prescriber. Results from all 4 auditing methods (automated query protocol and 3 manual audit methods) are stored in a master issue tracking database on Metabase.

### 2.3. Sample

The study included all GLP-1 RA prescriptions that were audited by the Eucalyptus clinical auditing team using any of its 4 methods between 1 April and 1 October 2023.

### 2.4. Procedures

Data were retrieved from the Eucalyptus clinical auditing team’s issue tracking database. Members of the team reviewed all identified errors to confirm their veracity. A csv spreadsheet was extracted for study investigators’ analysis.

### 2.5. Endpoints

The primary endpoint was the GLP-1 RA prescribing error rate of the Eucalyptus DWLS, which was calculated by dividing this total number of confirmed errors by the total number of prescriptions audited over the study period. Exploratory endpoints included an analysis of the distribution of errors across error types, error severity levels and the different auditing methods. The Eucalyptus severity matrix ranged from 1—‘low’ to 4—“never events” and were determined by the auditing team after a manual assessment the incident (Table 1).

### 2.6. Statistical Analysis

Descriptive data were presented in total numbers, mean scores and standard deviation figures. Chi-square tests were used to compare error rates across auditing types, gender, and GLP-1 RA type, whereby data were organized into ‘error’ and ‘no error’ columns of a contingency table. Point-biserial correlation tests were conducted to assess whether error rates were influenced by continuous variables such as patient age or body mass index (BMI). All analyses were performed on RStudio, version 2023.06.1+524 (RStudio: Integrated Development Environment for R, Boston, MA, USA).

## 3. Results

Between April and October 2023, 64549 GLP-1 RA prescriptions were issued via the Eucalyptus Australia DWLS. Baseline characteristics of the patients who received these prescriptions are presented in Table 2.

Of these 64,549 prescriptions, 29,595 were flagged as high-risk errors or ‘never events’ by Eucalyptus’ automated query system. After reviewing these queries, auditors confirmed that 1074 were prescribing errors. Over the same period, 2800 random audits were conducted, detecting 115 errors; 490 ad hoc audits detected 117 errors; and 4438 new prescriber audits detected 348 errors. Thus, the highest proportion of errors was detected via the automated query method, followed by new prescriber audits, ad hoc audits and random audits (Figure 2).

To calculate the final prescribing error rate, authors had to add the numerators and denominators from the four discrete auditing types. In total, 37,323 GLP-1 RA weight-loss prescriptions were audited, from which 1654 errors were detected, representing an error rate of 4.4%. The highest error detection rates were observed in ad hoc audits and new prescriber audits, at 23.9% and 7.8%, respectively (Table 3). A total of 87 high-risk or ‘never event’ errors were identified, with a disproportionate number observed in ad hoc audits (7 out of 490 audited consults (1.5%)). The most common errors were a failure to deliver sufficient safety counselling (49.15%) and inadequate investigation of a potential contraindication (30.29%).

The results of a multivariate binary logistic regression revealed that, controlling for all other variables, prescriptions for patients who were from the highest baseline BMI category (≥40 kg/m^2^) were over 25% more likely to contain errors than those for patients from each of the lower BMI categories (*p* < 0.001) (Table 4). This regression analysis also found that the higher error detection rate observed in ad hoc audits relative to the other three audit types was statistically significant (*p* < 0.001). Odds ratios in Table 4 were calculated by exponentiating the corresponding regression coefficient.

## 4. Discussion

This is the first study to report prescribing error rates in a real-world GLP-1 RA-supported DWLS. Against the backdrop of a global obesity epidemic, rising care access challenges and widespread knowledge of GLP-1 RA efficacy, these services are becoming increasingly important. Our retrospective analysis observed a low prescribing error rate in the Eucalyptus Australia DWLS over the study period. It was also found that most errors pertained to insufficient safety counselling and inadequate investigation of potential contraindications.

As discussed in the introduction, existing prescribing error rate data do not lend themselves to a clear comparison with the rate measured in this study. Although the 4.4% error rate detected in the Eucalyptus DWLS appears low relative to the figures reported in previous Australian and international studies, setting disparities are too significant to even suggest a loose benchmark. Furthermore, very few prescribing error rate studies have been carried out in the past five years, a period in which technological advances have likely changed e-prescribing functionality to a significant degree. As a result of this knowledge gap, the regulatory landscape for DWLSs in Australia and other countries remains somewhat underdeveloped. For example, DWLS-specific regulation in Australia remains confined to advertising standards and GLP-1 RA compounding [33,34], while in the UK, guidelines stress that GLP-1 RAs should only be prescribed to weight-loss patients as an adjunct to multidisciplinary lifestyle therapy without any additional safety protocols around prescribing [5]. Therefore, the prescribing error rate observed in this study lays an important foundation for ongoing research on DWLS safety and the eventual establishment of regulatory standards for digital GLP-1 RA prescribing.

The study’s secondary measures also generated several novel revelations. The distribution of error types was largely inconsistent with the existing literature on digital prescribing [23]. Whereas incorrect dose or incorrect medication concentration have been reported as the most common error types in other community settings, nearly 80 percent of errors in the Eucalyptus DWLS were related to insufficient safety counselling or inadequate assessment of potential contraindications. Again, these disparities could reasonably be attributed to the unique features of GLP-1 RA-supported DWLSs. It is possible that the relative novelty of GLP-1 RA use in obesity care explains the higher frequency of counselling- and contraindication-related errors. In contrast, the limited scope of GLP-1 RA dosing may explain why incorrect dose and titration schedule errors were relatively infrequent. The discovery that ad hoc audits had the highest correct error detection rate of the four audit types was arguably unsurprising, given that they were based on internal suspicion. However, the finding that all audit types detected a significant number of errors indicates that a multi-method auditing approach has merit in real-world GLP-1 RA-supported DWLSs. The fact that only 4% of errors detected by automated queries were confirmed as true errors can be explained by the queries’ inability to capture follow-up questions. Eucalyptus may consider improving the efficiency of this automated query model. Finally, the study observed that patients from the highest BMI category (40+ kg/m^2^) were overrepresented in the Eucalyptus service’s prescribing error data. This result could possibly be explained by the group’s higher rate of comorbidities, but further investigation would be required to draw any strong conclusions.

### 4.1. Public Health Implications

The study’s findings could have various implications for public health systems. Firstly, the reported prescribing error rate may be considered together with the recent study on DWLS dispensing errors [16] as preliminary evidence of the potential of GLP-1 RA-supported DWLSs to deliver safe obesity care. Dissemination of these findings will hopefully encourage researchers to conduct comparable investigations on other DWLSs and contribute to the development of national medication error standards for digital chronic care services. Secondly, the distribution of error types in the Eucalyptus DWLS illuminates the importance of educating GLP-1 RA prescribers of the medications’ contraindications and general safety profile. They also suggest that an alert system could be implemented to block clinicians from prescribing GLP-1 RAs if they have not provided sufficient counselling or conducted a thorough assessment of potential contraindications. Thirdly, the range of Eucalyptus DWLS error types and the effectiveness of all four auditing methods in capturing errors highlight the utility of the service’s multi-method auditing approach. Public health systems might consider developing a set of clinical governance standards for DWLSs and other digital prescribing services, which contain guidelines around auditing protocols. The Eucalyptus DWLS method could be used as a starting point for the development of such standards. Finally, the discovery that patients from the highest BMI category (BMI ≥ 40 kg/m^2^) experienced a disproportionately high error rate adds to the knowledge that the cohort also tends to lose less weight in GLP-1 RA-supported DWLSs [30]. Stakeholders might interpret this as further evidence for the need to dedicate additional resources to this high-risk group.

### 4.2. Limitations

The study has some limitations; firstly, only the prescriptions that were captured by Eucalyptus’ four auditing methods were included in the study, which accounted for 57.82% of the total number of GLP-1 RA prescriptions issued over the study period. Although it is unlikely that the combination of methods would have missed a significant number of errors given their collective comprehensiveness, the possibility of oversight and, thus, sampling bias cannot be categorically ruled out. Specifically, the random audit method would have likely undersampled various error types given the nature of its prescription selection, and the automated method could have possibly missed errors that stemmed from unique inputs in the Eucalyptus data repository. Secondly, the Eucalyptus auditing team (doctors and pharmacists) was responsible for retrospectively verifying prescription errors and thus may have exhibited various biases toward the company. Thirdly, the cohort of patients who were prescribed GLP-1 RAs through the Eucalyptus DWLS contained a disproportionately high number of women and Caucasians and was thus not representative of Australian society. Fourthly, the study only assessed errors over a period of six months, which may not be long enough to capture the full scope of DWLS prescribing practices. Finally, the study’s findings may not be generalizable to other GLP-1 RA-supported DWLSs, as Eucalyptus could feasibly have very different prescribing and auditing protocols to other modern DWLSs. Other services might also serve populations with dissimilar baseline data to those reported in the Eucalyptus cohort, such as a higher proportion of patients in ≥40 kg/m^2^ BMI categories.

### 4.3. Future Research

The logical follow-up to this investigation would be to conduct comparable analyses of other GLP-1 RA-supported DWLSs to assist in the establishment of a medication error benchmark for the industry. Investigators should consider reporting GLP-1 RA prescribing and dispensing error rates in face-to-face services, along with detailed case studies of DWLS clinical governance protocols and clinical follow-up to prescribing errors of medium and high severity.

## 5. Conclusions

This study generated vital foundational knowledge on prescribing safety in GLP-1 RA-supported DWLSs. These services are becoming increasingly important in the global fight against obesity but remain severely understudied. This study’s findings on the prescribing error rate, error types and auditing methods of Australia’s largest DWLS complement a recent study on the service’s dispensing error rate and lay a foundation for ongoing safety investigations of DWLSs. Although GLP-1 RA-supported DWLSs have the potential to contribute to a fundamental shift in global obesity rates, much more research is needed to support the development of safe care models.

## Figures and Tables

**Figure 1 healthcare-12-02093-f001:**
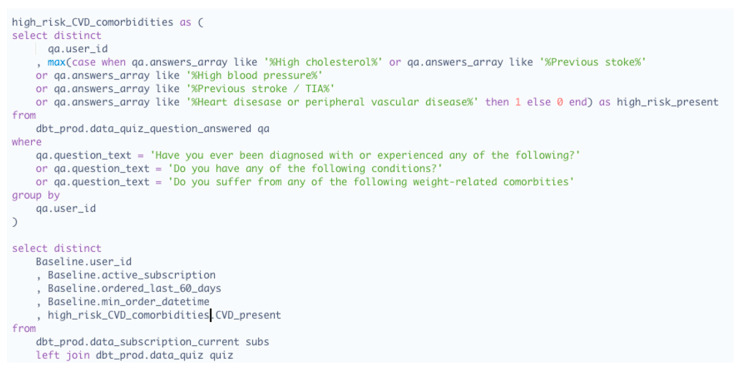
Example query for automated prescribing error auditing method, using Metabase’s SQL server.

**Figure 2 healthcare-12-02093-f002:**
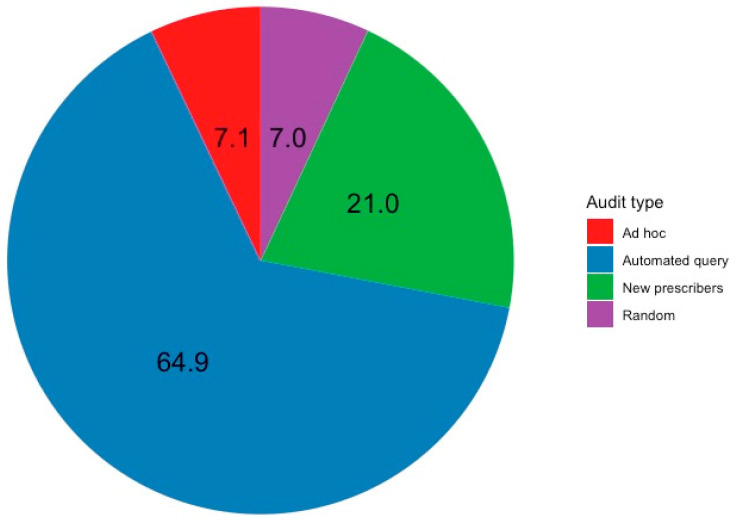
Percentage distribution of detected errors by auditing method.

**Table 1 healthcare-12-02093-t001:** Eucalyptus prescribing error severity ratings.

Severity Rating	Description	Example
4—Never event	Death or likely permanent harm that is not reasonably expected as an outcome of healthcare/weight-loss treatment	Patient hospitalized for attempted suicide from being prescribed Liraglutide with a known serious mental illness, not considered by the prescriber.
3—High	Temporary major harm or permanent consequences that are not reasonably expected as an outcome of healthcare/weight-loss treatment	Patient prescribed GLP-1 RA dose that exceeds Australian Therapeutic Goods Administration guidelines.
2—Medium	Minimal/minor harm that is not reasonably expected as an outcome of healthcare/weight-loss treatment	Patient selected “Eating disorder (anorexia, bulimia)”. No further assessment or clarification was requested from the prescribing doctor.
1—Low	Narrowly avoided harm	Clinician prescribes a patient Semaglutide without confirming they have counselled them on possible side effects.

**Table 2 healthcare-12-02093-t002:** Baseline characteristics of patients whose GLP-1 RA prescriptions were audited during the study period.

**Demographic information**	**Mean (SD)**
Age	43.79 (±7.92) years
**Gender**	**Number (%)**
Female	18,686 (76.70)
Male	5676 (23.30)
**Ethnicity**	**Number (%)**
Caucasian	20,049 (82.30)
Asian including subcontinent	1502 (6.17)
Aboriginal or Torres Strait Islander	1339 (5.50)
Pacific Islander or Māori	793 (3.26)
Latino/Hispanic	508 (2.09)
Other	171 (0.7)
**Clinical information**	**Mean (SD)**
BMI	33.92 (±6.05) kg/m^2^
Weight	98.79 (±18.94) kg

**Table 3 healthcare-12-02093-t003:** Prescribing error details.

**Safety audits by audit type—no. (% of total prescriptions audited)**		
Automated query	26,821 (71.9)	
Random	2800 (7.5)	
Ad hoc	490 (1.3)	
New prescribers	4438 (11.9)	
Total	37,323 (100)	
**Errors by audit type—no. (% relative to audit type)**		
Automated query	1074 (4.0)	
Random	115 (4.1)	
Ad hoc	117 (23.9)	
New prescribers	348 (7.8)	
Total	1654 (4.4)	
**Error severity—no. of errors (% of total errors)**		
4-Never event	9 (0.5)	
3-High	78 (4.7)	
2-Medium	778 (47.0)	
1-Low	797 (48.2)	
**Highest risk (rating 4 and 5) errors by audit type—no. (% of total errors of this severity)**		
Automated query	61 (70.1)	
Random	6 (6.9)	
Ad hoc	7 (8.0)	
New prescribers	13 (14.9)	
**Error type—no. (% of total errors)**		**Examples**
Failure to provide sufficient safety counselling	813 (49.15)	Patient is planning to conceive following the Eucalyptus program. Physician did not discuss safety considerations.
Inadequate investigation of a potential contraindication	501 (30.29)	Patient has a history of kidney stones. Physician did not confirm whether they were removed.
Incorrect dose	187 (11.31)	Patient was prescribed 1 mg of Semaglutide as an initial dose.
Titration schedule errors	153 (9.25)	Patient was prescribed 1.2 mg of Liraglutide for an additional week without explanation

**Table 4 healthcare-12-02093-t004:** Logistic regression model of predictors of Eucalyptus prescription errors.

Covariate	N	Odds Ratio	95% Confidence Interval	*p*-Value
**Age**	24,362	1.00	(0.996, 1.003)	0.784
**Ethnicity**				
Caucasian	20,049	Reference	Reference	Reference
Aboriginal or Torres Strait Islander	1339	0.917	(0.731, 1.136)	0.442
Asian including subcontinent	1502	1.033	(0.843, 1,254)	0.747
Pacific Islander or Māori	793	1.091	(0.827, 1.411)	0.523
Latino/Hispanic	508	0.874	(0.597, 1.232)	0.464
Other	171	0.601	(0.270, 1.147)	0.162
**Gender**				
Female	18,686	Reference	Reference	Reference
Male	5676	0.996	(0.888, 1.116)	0.95
**BMI category**				
≥40 kg/m^2^	4547	Reference	Reference	Reference
35–39.99 kg/m^2^	6614	0.696	(0.605, 0.801)	<0.001 ***
30–34.99 kg/m^2^	7408	0.773	(0.676, 0.884)	<0.001 ***
27.5–29.99 kg/m^2^	5793	0.761	(0.659, 0.877)	<0.001 ***
**Audit type**				
Ad hoc	490	Reference	Reference	Reference
Automated query	29,595	0.104	(0.086, 0.128)	<0.001 ***
New prescribers	4438	0.163	(0.131, 0.204)	<0.001 ***
Random	2800	0.098	(0.075, 0.128)	<0.001 ***

Note: *** *p*-value < 0.001.

## Data Availability

The data presented in this study are available from the corresponding author on reasonable request.

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
