# Peer review of "GLP-1 RA Prescribing Errors in a Multidisciplinary Digital Weight-Loss Service: A Retrospective Quantitative Analysis"

_healthcare, 2024, doi:10.3390/healthcare12202093_

Round 1

Reviewer 1 Report

Comments and Suggestions for Authors

1.     The study is confined to a single service (one specific DWLS in Australia (Eucalyptus)) which makes it difficult to directly generalize the results to other digital health services.

2.     The study could have explored ways to refine or optimize the automated system to reduce false positives, as it showed a high false positive rate.

3.     The study's short period (only six months) is not sufficient to capture the full scope of prescribing patterns as prescribing practices and error rates could fluctuate over longer periods, especially when new prescribers join or existing protocols change.

4.     Lack of follow-up on patient outcomes related to the prescribing errors makes it hard to fully grasp the real-world impact of such errors.

5.     The study does not address how prescriber experience correlates with safety, as it does not differentiate between prescribing errors made by more experienced clinicians versus those made by new prescribers. Such data would help develop targeted training and onboarding programs.

6.   The study relies on the audits to detect errors, this can lead to potential underreporting of errors that are only discovered when adverse events occur later.

Author Response

Comment 1: The study is confined to a single service (one specific DWLS in Australia (Eucalyptus)) which makes it difficult to directly generalize the results to other digital health services.

Response: Thank you for recognizing this limitation. We have now added the following two sentences to our ‘Limitations’ sub section: “And finally, the study’s findings may not be generalizable to other GLP-1 RA-supported DWLSs, as Eucalyptus could feasibly have very different prescribing and auditing protocols to other modern DWLSs. Other services might also serve populations with dissimilar baseline data to those reported in the Eucalyptus cohort, such as a higher proportion of patients in ≥40 kg/m2 BMI categories.”

Comment 2: The study could have explored ways to refine or optimize the automated system to reduce false positives, as it showed a high false positive rate.

Response: Thank you for noticing this high false positive rate. We feel that an exploration of ways to optimize the automated error detection method was beyond the scope of this study. However, we have now outlined the method’s limitation of possibly overlooking unique data inputs in the ‘Limitations’ sub-section: “Specifically, the random audit method would have likely undersampled various error types given the nature of its prescription selection and the automated method could have possibly missed errors that stemmed from unique inputs in the Eucalyptus data repository.”

Comment 3: The study's short period (only six months) is not sufficient to capture the full scope of prescribing patterns as prescribing practices and error rates could fluctuate over longer periods, especially when new prescribers join or existing protocols change.

Response: Thank you for this valuable recommendation. We have now added the following sentence to the ‘Limitations’ sub-section: “Fourthly, the study only assessed errors over a period of six months, which may not be long enough to capture the full scope of DWLS prescribing practices.”

Comment 4: Lack of follow-up on patient outcomes related to the prescribing errors makes it hard to fully grasp the real-world impact of such errors.

Response: Thank you for this excellent point. As we noted in the introduction and discussion sections, this is the first study to measure prescribing safety in a DWLS or comparable setting. Consequently, we prioritized a dedicated analysis of prescribing errors, as we felt that an assessment of clinical follow-up to prescribing errors would blur the study’s objectives given the amount of data that would generate. To clarify this we have added the following sentences to the study design sub-section: “Key ethical implications of the study were the potential for patient harm if prescribing errors were not addressed by Eucalyptus in a timely manner or if patient data was not adequately deidentified and secured. An assessment of clinical responses to prescribing errors was considered to be beyond the scope of this study. Eucalyptus accepts full responsibility of any GLP-1 RA prescribing errors from its DWLS that lead to patient harm.” And the following clause to the ‘Future research’ sub-section: “Investigators should consider reporting GLP-1 RA prescribing and dispensing error rates in face-to-face services, along with detailed case studies of DWLS clinical governance protocols and clinical follow-up to prescribing errors of medium and high severity.”

Comment 5: The study does not address how prescriber experience correlates with safety, as it does not differentiate between prescribing errors made by more experienced clinicians versus those made by new prescribers. Such data would help develop targeted training and onboarding programs.

Response: Thank you for this insightful suggestion. Unfortunately, clinician experience data was not available for the study period. However, Eucalyptus started collecting this data in May 2024 and therefore analyses of this nature will be available for future studies of the service.

Comment 6: The study relies on the audits to detect errors, this can lead to potential underreporting of errors that are only discovered when adverse events occur later.

Response: Thank you for this important comment. We have highlighted this as a limitation on lines 369 to 371 as follows: “Secondly, the Eucalyptus auditing team (doctors and pharmacists) were responsible for retrospectively verifying prescription errors and thus may have exhibited various biases toward the company.”

Reviewer 2 Report

Comments and Suggestions for Authors

Obesity has become a global concern. WHO alerts that "obesity is a serious public health problem that affects millions of people worldwide and causes diet-related noncommunicable diseases." Obesity is on the rise across the world, and medical professionals emphasize the importance of losing weight to treat and prevent chronic diseases such as diabetes and heart disease. Before exercise and diet-based weight loss were the main options, and it is often seen as a very long process, often resulting in slow progress and weight gain. Recently there has been an explosive growth in the weight loss-specific industry focused on line instead of obesity clinics that people. There have been many peptide-based weight loss treatments like semaglutide, etc. that focus on the GLP receptors and pathways that increase satiety. The use of glucagon-like peptide receptors has helped millions of people to help reduce obesity. There have been many safety concerns with cases that were fatal. Due to the safety aspect, these medications and treatments should be prescribed very cautiously, especially with digital weight loss programs, which often lack the safety aspect often used in in-person patient visits and clinics. It is important to have strict protocols to reduce risks and reduce prescribing errors. It is important to standardize and identify areas to reduce practicing errors. 

This study looked at retrospectively analyzing prescription errors among a large number of prescriptions. It adds an important data set to form better guidelines to avoid or limit prescription errors in a digital setting.

The abstract is hard to understand, and I suggest the authors work on better flow and making it easier for a reader to understand. Similarly, the methods sections are well written and explain the methods pretty clearly. 

Did the authors look at analyzing the error data based on the age, sex, educational qualification/awareness, and ethnicity? It would be interesting to see if the error rate is different for different groups within the total data set (age, male vs. female, educational level, etc.). 

It would be advantageous to add more information in the results sections for tables 3 and 4. 

The discussion is well written, and I applaud the authors for adding a section for limitations and future research. 

Overall, a good study looking at an important aspect for increasing safety in digital prescriptions. 

Comments on the Quality of English Language

Flow can be made better

Reviewer 3 Report

Comments and Suggestions for Authors

This manuscript investigated prescribing errors within a digital weight-loss service (DWLS) that used glucagon-like peptide-1 receptor agonists (GLP-1 RAs) as a supplement to lifestyle therapy. The study aimed to understand the types and severity of these errors in a real-world setting, focusing on Australia's largest GLP-1 RA-supported DWLS, Eucalyptus.

1. Main Question and Explicit Mention:

The main question addressed by the research is: What is the rate, type, and severity of prescribing errors within a real-world digital weight-loss service that uses GLP-1 RAs? While not explicitly stated in the abstract, the research question is clear from the introduction (lines 43-45) and the objectives (lines 90-93).

2. Originality and Relevance:

This study made a significant contribution to the field by:

  • Addressing a critical gap in the literature on digital prescribing safety: While several studies have investigated electronic prescribing errors in hospital and community clinic settings, research focusing on DWLSs, particularly those using GLP-1 RAs, is limited (lines 46-62). This gap is particularly important given the increasing popularity of DWLSs and the growing use of GLP-1 RAs for weight management.
  • Providing valuable real-world data: The study uses data from a large-scale DWLS, offering insights into actual prescribing practices and error rates beyond randomized controlled trials. This real-world perspective is crucial for understanding the complexities and challenges of prescribing in this emerging healthcare setting.
  • Analysing error types and severity: The manuscript goes beyond simply calculating the overall error rate and delves into the specific types of errors, their severity, and the contributing factors. This detailed analysis provides a more nuanced understanding of the risks associated with GLP-1 RA prescribing in DWLSs and informs targeted interventions.
  • Investigating the impact of different auditing methods: The authors analysed error detection rates across various auditing methods, highlighting the effectiveness and limitations of each approach. This analysis is particularly relevant for the development of robust auditing systems in DWLSs, ensuring comprehensive error detection and appropriate corrective action.

3. Similar or Related Literature:

While the authors have adequately cited existing literature, there are several relevant articles not mentioned that could be incorporated:

  • "Medication errors in e-prescribing: a systematic review" by Roumeliotis et al. (2019) [1]: This review provides a comprehensive overview of electronic prescribing errors, including their causes and potential consequences. This review could be referenced to provide a broader context for the study's findings on prescribing errors in a digital setting.
  • "Medication safety in Australia: A literature review" by Roughhead et al. (2013) [2]: This report summarizes the landscape of medication safety in Australia, highlighting the need for robust data collection and analysis. This study could be used to contextualize the importance of the current study within the broader Australian healthcare landscape and to emphasize the need for further research in this area.
  • "Electronic prescribing improves medication safety in community-based office practices" by Kaushal et al. (2010) [3]: This study provides evidence of the positive impact of electronic prescribing on medication safety in community settings. This study could be cited to compare the effectiveness of electronic prescribing in different healthcare settings and to support the authors' claims about the potential benefits of digital prescribing for medication safety.

4. Specific Improvements:

The authors could consider the following improvements to enhance the rigor and clarity of their study:

  • Clarifying the operational definition of "prescribing errors" (line 146): While the authors mentioned using a severity matrix (Table 1), a more detailed explanation of the criteria used for error classification is needed. Specifically, providing examples of each error type and how they were identified would enhance the study's reproducibility. For example, the authors could provide specific examples of "failure to provide sufficient safety counselling" or "inadequate investigation of a potential contraindication."
  • Providing further detail on the automated query protocol (line 147): The authors mentioned that these queries pull data onto thematic dashboards, but they should elaborate on the types of queries, their specific content, and how they trigger a manual review. This would allow readers to better understand the process of error detection and the potential for bias. For example, the authors could provide specific examples of the queries used, such as queries related to patient history, allergies, or contraindications, and explain how these queries identify potential prescribing errors.
  • Addressing the limitations of the random audit (line 196): The authors acknowledged that the random audit may miss certain errors, but they should further explain the potential for bias due to the randomness of the selection process. The authors could discuss the potential for oversampling or undersampling of specific error types due to the random nature of the audit.
  • Investigating the potential influence of patient demographics on error rates (line 216): While the study analysed the association between BMI and error rate, exploring the influence of other demographic factors, such as age and gender, could provide further insights into the potential for disparities in prescribing practices. For example, the authors could examine whether there are differences in prescribing error rates for patients of different ages or genders. This analysis could reveal potential bias in the prescribing process and inform interventions to improve prescribing safety.
  • Considering a more robust statistical analysis (line 178): Although the authors used chi-square tests and point-biserial correlation, more sophisticated statistical methods, such as logistic regression, could be used to explore the relationship between error rates, audit type, and patient characteristics. Logistic regression would allow the authors to control for potential confounding factors and to more accurately estimate the relative influence of different variables on prescribing errors. This analysis could provide a more nuanced understanding of the factors that contribute to prescribing errors and support the development of more targeted interventions.
  • Discussing the ethical implications of the study (line 97): The authors mentioned that the study was approved by the ethics committee, but they should also discuss any ethical concerns related to the retrospective analysis of patient data, particularly in the context of potential biases and the impact of error detection on patient care. For example, the authors could discuss the potential for patient harm if prescribing errors were not detected or if corrective action was not taken in a timely manner. They could also discuss the potential for privacy breaches if patient data was not adequately anonymized and secured.

5. Consistency of Conclusions with Evidence:

The conclusions are generally consistent with the evidence and arguments presented. The study demonstrates a low prescribing error rate within the Eucalyptus DWLS. However, the authors also highlight the prevalence of specific error types, such as inadequate safety counselling and investigation of contraindications, which suggests areas for improvement in the service's clinical governance practices.

The study did not explicitly address the potential influence of patient characteristics, such as BMI, on prescribing error rates; however, the authors acknowledge the overrepresentation of high-BMI patients in error data (line 268), setting the stage for further investigation.

6. Comments on Tables and Figures:

The tables and figures are generally well-presented and informative. Table 2 provides a clear overview of patient demographics and clinical information. Table 3 presents a detailed breakdown of prescribing error rates by audit type and severity level. However, the authors could consider:

  • Enhancing Table 3 by including the specific examples of errors for each category (line 178): This would provide a more comprehensive understanding of the error types.
  • Including a visual representation of the error distribution across different audit types (line 197): A bar chart or pie chart could further illustrate the distribution of errors across the four audit methods.

7. Caveats, Weaknesses, and Mistakes:

  • The authors might consider adding a section on the limitations of the study (line 293): While they briefly mentioned some limitations, a dedicated section would provide a more comprehensive overview of the study's potential shortcomings. For example, the authors could discuss the limitations of the study design, such as the retrospective nature of the data, the reliance on a single DWLS, and the potential for bias in error identification. They could also discuss the limitations of the study's sample size and the potential for sampling bias.
  • The authors could discuss the generalizability of their findings to other DWLSs (line 299): While the study focused on Eucalyptus, its findings may not be applicable to all DWLSs. The authors could discuss the potential for variation in prescribing practices and error rates across different DWLSs, which could be due to factors such as the specific patient population served, the model of care provided, and the training and experience of the clinical team.
  • The authors should provide more context on the specific limitations of the automated query protocol (line 147): While they acknowledged its limitations, they should further elaborate on the types of errors it might miss and why this could contribute to the observed discrepancy between errors detected and errors confirmed. For example, the authors could discuss the potential for the automated query protocol to miss errors related to patient history, allergies, or contraindications, which are not explicitly captured in the query.
  • The study's findings could benefit from a discussion of the current regulatory landscape for DWLSs in Australia and internationally (line 242): This context would further highlight the significance of the study's contribution to the field of digital prescribing safety and the need for robust standards. The authors could discuss the current regulatory framework for DWLSs in Australia and other countries, highlighting the challenges and opportunities for implementing best practices for prescribing safety.

References:

  1. Roumeliotis, N., Sniderman, J., Adams-Webber, T., et al. (2019). Effect of electronic prescribing strategies on medication and harm in hospital: a systematic review and meta-analysis. Journal of General Internal Medicine, 34, 2210-2223.
  2. Roughhead, L., Semple, S., Rosenfeld, E. (2013). Literature review: Medication safety in Australia. Australian Commission on Safety and Quality in Healthcare.
  3. Kaushal, R., Kern, L., Barrón, Y., et al. (2010). Electronic prescribing improves medication safety in community-based office practices. Journal of General Internal Medicine, 25, 470-478.

Round 2

Reviewer 3 Report

Comments and Suggestions for Authors

Glad with changes